# Cognitive Ergonomics of Assembly Work from a Job Demands–Resources Perspective: Three Qualitative Case Studies

**DOI:** 10.3390/ijerph182312282

**Published:** 2021-11-23

**Authors:** Matilda Wollter Bergman, Cecilia Berlin, Maral Babapour Chafi, Ann-Christine Falck, Roland Örtengren

**Affiliations:** 1Department of Industrial and Materials Science, Division of Design & Human Factors, Chalmers University of Technology, 412 96 Gothenburg, Sweden; matilda.wollterbergman@chalmers.se (M.W.B.); maral.babapour.chafi@vgregion.se (M.B.C.); 2The Institute of Stress Medicine, Region Västra Götaland, 413 19 Gothenburg, Sweden; 3Department of Industrial and Materials Science, Division of Production Systems, Chalmers University of Technology, 412 96 Gothenburg, Sweden; ann-christine.falck@chalmers.se (A.-C.F.); roland.ortengren@chalmers.se (R.Ö.)

**Keywords:** cognitive ergonomics, mental workload, manual assembly, work environment, cognitive performance, human factors, occupational ergonomics

## Abstract

In manufacturing companies, cognitive processing is required from assembly workers to perform correct and timely assembly of complex products, often with varied specifications and high quality demands. This paper explores assembly operators’ perceptions of cognitive/mental workload to provide a holistic understanding of the work conditions that affect cognitive demands and performance. While the physical loading aspects of assembly work are well known, most empirical literature dealing with cognitive/mental loading in manufacturing tends to examine a few particular aspects, rather than address the issue with a holistic system view. This semi-structured interview study, involving 50 industrial assembly operators from three Swedish companies, explores how assemblers perceive that their cognitive performance and well-being is influenced by a wide variety of factors within the context of mechanical product assembly. The interview transcripts were analysed using a priori coding, followed by bottom-up Thematic Analysis. The results indicate that a variety of systemic effects on assemblers’ cognitive performance can be classified as job demands or resources. Quite often, the absence of a resource mirrors a related demand, and “good assembly conditions”, as described by the interviewees, often re-frame demands as desirable challenges that foster motivation and positive feelings towards the work. The identified demands and resources stem from task design, timing, physical loading, intrinsic and extrinsic motivators, social teamwork and the product’s “interface” design. Despite organisational differences and conditions between the three companies that took part in the study, the results are largely consistent.

## 1. Introduction

To increase output and ship value to customers, manufacturing companies strive to achieve correct and consistent assembly of complex products under accelerating market demands for high-quality and timely delivery. A well-established concern for companies is high physical loading and its threat to quality [1,2,3]. While assembly work has long been (and remains) physically straining, little is known about the overall consequences of cognitive loading [4] on assemblers’ task performance, well-being and the speed and quality of production. Cognitive under- or overloading can cause a lack of focus or distraction on one hand, or overwhelm on the other [4]. This paper explores factors relevant to assembly operators’ perceptions of cognitive/mental workload in order to provide a holistic understanding of work conditions related to cognitive demands and performance.

Cognitive ergonomics, a well-established area within the discipline of Human Factors/Ergonomics (HF/E), aims at designing systems to support optimal human cognitive performance and well-being when performing tasks, in line with the International Ergonomics Association’s definition of HF/E [5]. This field studies human cognitive capacities (such as memory, perception, attention, decision making, problem-solving, etc.), how they interact with given elements of a work system (such as tools, controls, instructions and interfaces), factors that affect performance (such as mental workload, stress or distractions) and performance outcomes (such as time taken or degree of error when executing the task). The practice aspect of the discipline aims to influence the design of the work conditions that influence cognitive performance. The study reported in this paper aims to contribute knowledge towards improving cognitive ergonomics in manufacturing, where human workers perform manual assembly.

With regard to cognitive and/or mental loading in manufacturing contexts, several previous works bring up a variety facets to consider: how assemblers handle disruptions [6,7,8], overall process flow [9], effects of assembler age [10,11,12], interactions between physical and mental loading [8], assembler experience [13], boredom [14], efficacy and appropriateness of tools, materials and instructions [15,16,17], time pressure [18] and the effects of task complexity [1,13,19,20], just to name a few. This variety in the literature suggests that the scope of what constitutes mental or cognitive loading for assembly workers emerges from a systemic combination of any of these factors. It appears that assemblers do not simply deal with a succession of building tasks, but their work is also interlaced with interpreting new information, problem-solving, navigating social interactions, prioritising and learning on the job.

What may appear to be a simple matter of word choice between the concepts of *cognitive* vs. *mental* (work)load is not entirely straightforward in the literature. Given that the HF/E field is multi-disciplinary, is informed by a multitude of research traditions and has the aim of designing to optimize work, both terms are found in the literature, and sometimes appear to be used interchangeably. However, the term “mental workload” (MWL) is used predominantly in cognitive sciences and HF/E literature since the 1980s [4], and in the latter field tends to address the discrepancy between job demands and the (mental) resources available to the worker—based on the assumption that the worker is wholly dedicated to completing the task. Some theoretical HF/E treatments use one term to explain the other (e.g., [4]), or relate the terms to each other [18] in an attempt to frame the scope in terms of effects on worker performance.

In early HF/E contributions (circa 1980s), the mental side of human work tended to be approached simply as “information processing”; i.e., perceiving and interpreting sensory signals and deciding on a course of action based on that interpretation process [21,22]. Early HF/E literature has also been very performance- and safety-focused, often studying outcomes of task completion experimentally in terms of correct execution, time usage and risk of human error. Conclusively defining MWL based on the literature remains a complex challenge due to the multidimensionality, hypotheticality and wide variety of conceptualisations that have been used to elaborate and operationalise it; this elusiveness is exemplified by the ISO 10075 standard “Ergonomic principles related to mental workload“, which simply refrains altogether from defining MWL and refers to it as an “umbrella term”.

On the other hand, much of our fundamental understanding of the human brain at work stems from cognitive sciences and educational psychology. The latter tends to prefer the term “cognitive load”, and its perspective provides a contrast to the HF/E angle by *not* presuming that the person dedicates all mental resources to the (learning) task; rather, it discusses *whether* they apply their available resources actively, as a matter of choice. A widely adopted educational psychology model by Sweller, *cognitive load theory (CLT)* [23,24], segments cognitive load into *intrinsic* (directly related to task difficulty), *extraneous* (affected by surrounding external factors) and *germane* (the load placed on memory during the learning process when developing “schema” that allow comprehension). Later treatments of CLT [25] emphasize that certain types of cognitive load are experienced passively by the “learner”, while others cause them to actively invest effort in dealing with the task. 

One concept analysis and model by Van Acker et al. [26] (p. 358) builds on CLT and defines mental workload as “*a subjectively experienced physiological processing state, revealing the interplay between one’s limited and multidimensional cognitive resources and the cognitive work demands being exposed to*”. According to their model, mental workload arises in a situation where (1) a worker’s (personal) cognitive architecture meets (situational) cognitive task demands; (2) the worker spends cognitive resources and triggers a subjective experience that may be affected by emotions and moderated by specific employee- or work-related characteristics; and (3) this results in employee behaviour that may lead to desirable or undesirable levels of performance, as well as and other consequences that may affect the work performance, resilience and/or well-being positively or negatively. This definition implies that mental workload is a more all-encompassing, situationally focused concept, and that (task-focused) cognitive loading is simply a component thereof. A model by Hockey [27] supports that it would be wise to acknowledge the “*voluntary application of effort”* idea [25] to paced assembly-line scenarios, regarding performance; when carrying out tasks under stressful conditions, individuals may adapt to stress with an intentional allocation of effort, *or* by compensatory “latent decrements” to meet specific performance targets to the possible detriment of other goals, which may result in sub-optimal performance and fatigue. This echoes the ETTO principle (efficiency–thoroughness trade-off) described by Hollnagel [28], an idea that acknowledges how surrounding conditions may influence voluntary investment of effort towards completing a task successfully.

We propose that when studying the “assembler’s brain at work”, it should not be done simply in relation to isolated tasks, but with a regard to the assembly job as a whole. This endeavour requires a broad appreciation of how humans use their mental resources, both consciously and unconsciously. An industrial work system context expects correct and consistent performance from assembly workers at all times. For this reason, we have chosen to inclusively label the scope of our study inclusively as “*cognitive/mental*” demands and resources throughout the paper.

### 1.1. Theoretical Framework

The Job Demands–Resources (JD-R) model by Bakker and Demerouti [29] is used as a framework to explore and explain demands and resources pertaining to cognitive load in assembly work. According to the model, occupational strain and ill-health result from an imbalance between demands and resources that employees are subjected to. The model builds on two processes: (i) a health impairment process as the result of poor job design; and (ii) a motivational process coupled with job resources that are assumed to lead to work engagement and performance [29]. The JD-R model has been used to identify the impacts of work conditions on employee and organisational outcomes such as burnout and engagement [30,31], and absenteeism and turnover intentions [32].

The model has been used to investigate work conditions in various contexts; for example, call centres [32,33], caregiving environments [34,35], flexible offices [36,37] and teaching [38]. It has also been applied to manufacturing and assembly contexts; for example, Demerouti et al. [39] themselves carried out a JD-R study partially involving industrial workers, focusing on exhaustion and disengagement (burnout). In [40] (p. 3010), the model was applied to lean manufacturing, finding that, e.g., provision of training, boundary control and feedback were crucial “*for promoting engagement and reducing exhaustion”*. Elsewhere, the JD-R model was used to examine workers’ intentions to retire early; showing that “*the relationship between job resources and work enjoyment was stronger among blue-collar workers”* than for white-collar workers [41] (p. 61). Job demands are divided into physical, psychological and emotional, while resources can be support, autonomy and feedback [29]. Every occupation may have its specific and contextual demands and resources.

### 1.2. Aims and Research Questions

This interview study aims to explore and map how assemblers perceive that work conditions influence their cognitive/mental workload and performance, from both a hindering and a facilitating perspective. In line with JD-R, we focus on mapping the demanding factors and resources within the assembly context, stemming from the design of the product as well as the social and organisational aspects of the production setup. The research questions are: What are the contextual demands that influence cognitive workload and performance in assembly work?What are the contextual resources that influence cognitive workload and performance in assembly work?

### 1.3. Delimitations

According to Bakker and Demerouti [29] (p. 312), “*Job demands refer to those physical, psychological, social, or organizational aspects of the job that require sustained physical and/or psychological (cognitive and emotional) effort or skills (…).*” Due to the primary focus of this paper’s results being assemblers’ cognitive/mental workload and performance, we deliberately refrain from elaborating the physical aspects of demands and resources (although these are normally included in the JD-R model) as they would detract from the paper’s focus on cognitive and mental demands/resources.

## 2. Materials and Methods

Our study design and procedure is reported using the Consolidated Criteria for Reporting Qualitative Studies (COREQ) checklist [42], based on [43]. To investigate assemblers’ work conditions with respect to cognitive workload, a multiple case study approach was taken. Case study design is appropriate when (1) the aim is to understand complex interrelations between studied phenomena; (2) the research work aims to understand thick descriptions that represent different perspectives; and (3) the research has little control over studied events but is interested in naturally occurring variability [44,45]. Multiple-case study approaches, in particular, enable comparisons and allow for propositions to be made based on varied situations, contexts and perspectives, preventing the idiosyncrasies of findings from single cases [46]. Data collection, case descriptions and the analysis procedure of this multiple-case study are described in the following sections.

### 2.1. Data Collection

The study primarily consisted of semi-structured interviews with 50 individual manual assembly workers at three Swedish industrial manufacturing companies. The interviews were conducted individually at each company’s respective production site. A contact person (CP) who worked closely with the production organisation for each company was responsible for recruiting interviewees. The CP selected the assemblers who were permitted to leave the assembly line during their working day to participate in a scheduled interview for approximately 30–40 min. During the selection, the CP needed to fulfil the research team’s request to represent a variety of work experience levels, genders and ages; at all three companies, CPs intentionally selected participants based on their being able to speak Swedish. To minimise interference of assembly work as much as possible, the research team accepted the company’s selection of participants, and completed all interviews during the same day or two consecutive days.

Each interview was semi-structured, using a flexible interview guide (Appendix A). The questions were formulated to probe individuals’ work experience, understanding of their work tasks, what they perceived as enabling or hindering their work and how the products or the production tasks and work environment affected their ability to perceive signals, recall information from memory, solve problems and make decisions. In this paper, references to specific interview questions are labelled with the letter Q and the corresponding question number in Appendix A. For example, Q-12 corresponds to the question “*12. What makes your work flow well?*”.

### 2.2. Cases and Participants

Three industrial companies participated in the study, which we refer to as Case A, B and C, respectively. Two of these were multi-national vehicle producers with production plants in several countries, while the third was an automotive component manufacturer specialised in one very particular functional component. All companies volunteered to provide participants from their own production plants situated within Sweden, meaning that all organisations obeyed the same work environment laws and regulations. Table 1 shows the overall demographics of our sample. The sample should be compared with the overall gender distribution of the automotive assembly sector, which according to Statistics Sweden [47] was 20% women and 80% men (13,000 persons in total) at the time the interviewees were recruited. The sample also covers a span of prior work experience and age; the national average age for women was 34 years, and for men 39 years [47].

### 2.3. Data Analysis

The overall process for data analysis is shown in Figure 1. All interviews were audio-recorded and transcribed verbatim. For reporting purposes, selected quotes were translated from Swedish to English by a bi-lingual author. The transcript materials were thereafter segmented by categorising the answers into thematic “clusters” (see Figure 2). These clusters of specific answers were then subjected to bottom-up thematic analysis [48] to identify emergent themes. These identified themes were iteratively discussed in the research team (see Figure 1) and organised under the pre-established categories from the Job Demands and Resources model [29] (except for physical aspects, see Section 1.3). 

The results were analysed for each case company in turn, but the results are reported for common themes found in all three cases (exceptions where differences between them are reported are stated as such). Where *quotes* are reported, we sometimes truncate using *(…)* to indicate skipped text, and sometimes the subject or object being referred to is indicated in box parentheses ([“*subject*”]) to clarify. On occasions where multiple interviewees expressed a similar view, we attribute the quote with interviewee code names as follows, with the first mentioned being the original source of the quote: “*Quote*” *(A-#; B-#).*

## 3. Results

The results are divided into two main sections, providing an overview of the identified *demands* and *resources* in assembly work. Each section chiefly presents results that were in common between the three cases, and highlights interesting findings specific to a particular case, to illustrate their different conditions.

All interviewed assemblers in the three cases, with few exceptions, worked at paced assembly lines with a specified “takt time”, which limited how long they could spend carrying out any sub-assembly task. For most, the working day consisted of several time-limited placements (rotations) at a particular workstation, where they would perform assembly work from 20 min to a couple of hours, before switching to another station. In Case C, most products were very similar and produced in batches, while in Cases A and B each individual product (vehicle) came with a custom specification that required assemblers familiarize themselves with how to assemble that particular customized product. This could involve selection of components and materials, sometimes dealing with packaging, lifting and/or moving objects from nearby material racks to the assembly, and attachment of components to the product using one or more tools. The components and materials could sometimes be very heavy, occasionally requiring the use of lifting devices or pneumatic tools. As mentioned earlier in Section 1.3, the paper mainly focuses on the psychological, social and organisational aspects of JD-R, and will discuss physical demands and resources at a later stage.

Despite the interview focus on cognitive/mental loading, the interviewees frequently emphasised that they considered assembly work to be much more physically than mentally straining. Most (trained and accustomed) assemblers felt that assembly was mostly manageable, and that the main challenge was exposure to heavy or repetitive physical loading, leading to fatigue and cumulative injury, alongside a lack of control over the timing of the work tasks (since all interviewees worked at paced assembly lines). Most assemblers also associated the (Swedish) word “ergonomics’’ chiefly to physical demands; to mitigate this, the interview questions were purposely designed to probe for and focus on specific aspects of cognitive/mental workload (see Appendix A).

### 3.1. Demands in Assembly Work

The identified work demands were organised into (i) Cognitive demands; (ii) Emotional demands; (iii) Social and organisational demands; and (iv) Individual pre-conditions (related to demands).

#### 3.1.1. Cognitive Demands

The different factors related to cognitive demands are illustrated in Figure 3. A noticeable category comprised time pressure-related aspects of the work, in terms of pace, duration and variation (or lack thereof). One of the most predominant cognitive demands on assemblers was their awareness of the decreasing time left, in most cases emphasised by a visible clock counting down the time to the start of assembling the next product. Cycle times at paced workstations varied between approximately one and a half to seven minutes, indicating varying degrees of task repetitiveness. Particularly novice assemblers viewed this countdown as a source of stress and a distraction from focusing on the work at hand, as they were looking at the time more often in order to match expectations. In Cases A and B, the occurrence of unusual product variants with unexpected or unfamiliar assembly requirements were mentioned as having a large impact on “*time left*”. The feeling of running out of time could cause one assembler to work more carelessly: “*When you’re behind (…) the mistakes come. Because you more or less throw in the screws and washers. Whether or not it’s the wrong way round (…) I have to hurry on to the next one.*” *(A-12)*. Time pressure is also related to production system disturbances, for example machine failure or other stoppages. The resulting delay could cause great frustration. Interviewees from Case C reported that they were motivated to “catch up” with the tempo after problems in the line production had occurred. Another assembler mentioned the occasional need to measure components like beams to ensure tolerances: “*Sometimes we have to measure beams, and that’s super complicated (…) you also have to build all the other stuff too with the clock ticking. (…) It’s a thing you don’t want to do, start calculating math while you’re supposed to be building*” *(B-08)*.

In a worst-case scenario, activities that assemblers would need to fit into the available cycle time could include interpreting instructions, handling missing or faulty components, dealing with assemblies with poor fit, addressing queries from colleagues (or extending queries of their own) and generally needing to be quick-thinking and aware of what was coming next: “*You always need to stay one vehicle ahead, in your mind*” *(B-11)*; all this while sometimes handling heavy physical loads. Assemblers also mentioned that some degree of rapid *decision-making* was occasionally involved in the assembly, due to variability from the material and tools: “*you get a piece that’s just borderline on whether it’s wrong, and you have to stand there thinking ‘should I do something about this, or not?’*” *(C-05)*.

Work-environmental factors such as noise, blinking lights, etc., could also contribute to assemblers’ feeling “stress” and fatigue: “*you get tired in the body and the head because of all the noise*” *(B-13)*. In Case C, some such signals or tasks could come from parallel processes that assemblers were expected to monitor at the same time as their assembly tasks: “*So it’s hard enough to keep your focus on a station when you know that at any time, this other station might start beeping. Sometimes it might not even beep, so you have to turn around and check, remember, to check that it’s running. That can be mentally demanding*” *(C-01)*.

Precision-demanding work was another product-related factor highlighted by the interviewees in Cases A and B as mentally taxing, due to requiring thoroughness under time pressure. For example, components requiring high precision risked being mounted at a skewed angle, or otherwise failing quality criteria that would require extra effort and time to correct afterwards. This problem at the task level could stem from ambiguous component design, faulty materials or too-high tolerances that were difficult for human assemblers to live up to without some kind of quality-assurance guide or tool: “*when a connector or a screw or something that fits in several places (…) can end up wrong and when there’s so much to keep track of, it’s easy to make a mistake*” *(A-01)*.

Interviewees emphasised that unclear work instructions can be very demanding: “*It can be hard to know why a pipe should be rotated a certain way, because it’s not always evident from the instruction*” *(B-14).* It was also mentioned in all three cases that the work instructions could be incorrect and contain an enlarged amount of information that is demanding to read: “*There’s a lot (…) that isn’t right. There’s a lot of stuff to remember and rules-of-thumb (…) There’s also a lot of information like (…) someone writing ‘Note: assemble this’. But texts like that (…) you stop seeing them after a while, so they just add to the disorder*” *(B-08)*.

In Case A, the participants described “red stations” involving several demanding work tasks. These require additional help from another assembler, and no one is meant to work there too long. Workstations with too much or too little task variety that are poorly balanced could result in over- or underload: “*sometimes it can be very calm and very monotonous because it is a very easy sequence (…) then you become rather understimulated (…) but when there is a very high sequence things can go the other way, meaning that it’s too much (…) we have stations that are red, where we are not allowed to stay too long*” *(A-09)*.

Something that set Case C apart was that the complexity of their product was lower than those of the other two, so most assemblers from there reported that they found the cognitive load manageable. As one assembler described it: “*I [have to] think quite a lot, but I don’t find it particularly straining. It’s just a responsibility, that I should do my part correctly. And instead of doing it wrong, I’d rather think about it for two seconds extra*” *(C-12)*. In extension, too-low task variation could also lead to negative attitudes towards the work: “*Some days you think, ‘Well, here I am doing the same thing again’*” *(C-04)* and “*You do the same things every day. Every second almost*” *(A-08; A-10)*.

#### 3.1.2. Emotional Demands

The interviewees reported certain situations that involved emotional demands. Although overlapping somewhat with previously mentioned cognitive demands, interviewees reported feelings of frustration or irritation after experiencing system or component failures that they had no control over—this frequently had consequences like time delays and increased need for adjustments, staying overtime and additional physical and mental effort. Particularly, equipment failures seemed to cause frustration, since the assemblers would often have to wait for repairs before being able to resume work: “*When machines and stuff don’t work like they’re supposed to, (…) sometimes it takes a good while before they get fixed. And you have to stand there and fight with it. It gets pretty straining after a working day*” *(C-05)*. Despite lacking control, assemblers still experienced a sense of needing to catch up. Alternatively, they anticipated needing to stay for longer hours or work on weekends to meet production targets, a common source of dismay. Similar frustrations could come from assembled products not passing quality checks, which routinely required adjustments.

The products themselves could also cause emotional responses, particularly when their assemblies were predictably difficult and effort-demanding, with several of them occurring in succession. One assembler in Case B said that “*It depends on whether you get troublesome vehicles (…) last week was a really [expletive] day (…) we had really difficult vehicles and you were behind all the time, and found out that you made a mistake five vehicles ago (…) ‘how the [expletive] did that go wrong? How the [expletive] did I manage to do that? It can’t be right’. You go into denial (…) And then you try to concentrate even harder, even though you know another [expletive] car is coming again. So those days, you absolutely just want to go home as soon as possible*” *(B-12)*.

Assemblers also mentioned the emotional toll of combined physical and mental fatigue after long working days. This was particularly tangible if they were asked to stay overtime after a work shift. Some felt additional frustration at knowing that certain tasks would regularly require them to handle very heavy physical loads. It appears that due to the predominantly physical nature of the work, it was hard for the interviewees to disregard its effect on mental performance entirely when they were asked about the work being (cognitively) strenuous. The body-related aspects came up as dominant in their experience of the work.

One assembler described how some materials and components would not always fit well into the assemblies with narrow tolerances, leading to frustration and extra physical efforts to force them in: “*they can be pretty badly fitted, so you really have to bang on and bend them (…) they’re hopeless and they give you trouble*” *(B-08)*. The same assembler described a variety of sources of frustration and confusion, e.g., unplanned staffing changes, material being kitted incorrectly and special-operations assemblers showing up to workstations without warning and imposing their notes and instructions (B-08).

Another aspect worthy to note was how frustrations could arise from lack of communication: “*If the communication isn’t good, that you don’t get informed in good time so you can’t prepare yourself—that can be frustrating” (B-17).* Some frustrations could also arise from colleagues getting in each other’s way particularly when “working up” (working faster than the assigned pace): “*That person hasn’t understood that, or why, they cause irritation. It’s because the other person has to wait, even though their time hasn’t started. This makes some people really mad while others don’t care. But of course, I can get annoyed too if someone doesn’t move and makes me wait*” *(B-12)*.

Similarly, frustration could arise from a perceived lack of understanding from other parts of the production chain: “*It’s frustrating (…) when those who decide don’t listen to workers. (…) Often it’s a seller who has said [to the customer], ‘you can have that in your vehicle, no problems’, but then they’ve already manufactured a cable harness. Then there aren’t any cables for that button, and then it becomes my problem to add those cables*” *(B-03)*.

#### 3.1.3. Organisational Demands

Demands resulting from the organisation, scheduling and design of work were exemplified in several ways (Figure 4). In all three cases, it was common practice to lend operators from other lines, in order to cover gaps and provide enough staff at each line. However, this solution required a lot of effort to uphold a good workflow and affected group dynamics negatively: “*if we get someone from another line, who (…) hasn’t been in our group and doesn’t know how we sync, then it gets a bit choppy*” *(C-04).* Sometimes inexperienced staff could add a limitation that affected the variation for the entire team: “*when people are sick (…) we staff with people [from the temporary staff pool] who know several different departments, so they most often only know one or two stations within our department. So that person gets locked in, and we can’t rotate around, so maybe everyone has to stand still. This causes (…) [people to get] irritated*” *(B-16)*.

Some interviewees who had experienced being “lent out” to an unfamiliar station described that it could be demanding (but stimulating) to learn new things (B-08), while others felt that they were learning new things constantly (B-03, B-07, B-13), partly because of occasional changes in operations. Getting accustomed to the new tasks often involved them asking more questions of surrounding colleagues to verify that they were assembling correctly.

Regarding learning and onboarding, interviewees appreciated standardised routines and more practical rather than theoretical approaches for learning new assemblies; however, in Case C, one interviewee who had been working night shift during their onboarding process described how their particular introduction had suffered from a great lack of structure and coordination, which made it very hard for his group of novices to learn: “*[the supervisors] were going to teach us and were running between us (…) they were very skilled, but those two weren’t very coordinated. So one came and said, ‘this is how you do it’, and then after two days the second one came saying ‘what the [expletive] are you doing, you can’t do that!’*” *(C-08)*.

A reported practice was to standardise work tasks and reduce them to a shorter sequence with fewer steps, to ensure quality and to ease the onboarding process for new staff. However, it had an adverse effect as the operators’ motivation decreased and they felt disengaged towards their work instead. For example, one experienced assembler said that they felt “*limited with capital letters (…) when you’ve worked as long as I have (…) we [previously] had a completely different kind of assembly. So, I can’t say it’s gone in the right direction, certainly not (…) What I miss is the whole picture, now that I just do one part. You don’t get the same holistic (…) view, so to speak*” *(A-13).* Some interviewees felt that there were limited opportunities offered at their company to develop new competencies on the job.

Unexpected events and disturbances were a frequently reported issue that assemblers said affected the workflow. Several assemblers felt that it would be desirable to have better preparation and time planning to handle disturbances, and to reduce risks for unexpected events. Some assemblers stated that a combination of having enough time to complete the task, and to have been given enough time to become well-trained to properly perform the task (particularly newly introduced ones), were conditions that supported a good flow in their work.

Participants from all three cases reported that there were gaps in their routines for ending a shift, which delayed and increased their efforts towards preparing their shift start-up: “*If there’s been an issue (…) they can’t focus on how to turn over to us (…) it’s the same for us. If we have a problem (…) at the end of the day, then it might not give the right starting conditions for those who start in the evening. We might leave it a bit badly done because we haven’t been able to focus on it*” *(B-15); “They care about doing their thing, but not about the one coming after. Sometimes you have to refill your own material, and then it takes a few minutes before I can start*” *(C-12)*. Shift-work itself was recognized by participants in all cases as taxing and unfavourable to safety and quality: “*Shift-work creates tiredness and can lead to assembly mistakes or injuries*” *(A-08).*

Regarding the design and operation of the physical workplace, participants in all three cases described the effects of when material was not delivered on time or with inadequate quality, placing demands on staff: “*Materials handling to us doesn’t work very well, so we have to hunt for material*” *(B-04).* Other reported issues were: “*Faulty material*” *(C-09), “Things that don’t fit and you have to re-do stuff*” *(C-10), and “there isn’t as much material as the system says*” *(C-15)*. In Case C, about half of the interviewees stated that the equipment greatly impacted their experience of workflow; interviewee C-09 added that there were often machine failures to contend with. Several participants largely indicated that the ideal scenario was to only have necessary and functioning equipment to deal with, to avoid unnecessary demands. Another demanding factor at workstation level was when instructions were difficult to read and understand, or not updated.

#### 3.1.4. Social Demands

Regarding working with other people, the assemblers mostly emphasised how social interactions and teamwork were a strength, but demands appeared mostly due to unfavourable group dynamics, interdependencies between individuals’ performance and dealing with the presence of other people as a distraction.

Group dynamics was recognised by interviewees as influencing cognitive workload—particularly, a dysfunctional work dynamic could result in dissatisfying teamwork and lowered motivation to come to work: “*if there’s a bad vibe, of course it’s not as fun. It’s not as rewarding*” *(A-07).* In Case C, a lack of collaboration and communication could for example lead to colleagues not helping each other refill materials. One assembler (C-13) described that the social interaction of doing the right thing towards colleagues and making them feel at ease at work was key to achieving a good workflow.

In Case C, interviewees mentioned that having a predetermined production target for them as a team to constantly keep up with was demanding; often they would try to work faster than pace in order to reach their target. However, different assemblers had different capabilities to work faster than pace, which could contribute towards a dysfunctional group dynamic: “*some people here feel that, ‘yeah but let’s reach our goal anyway’. And that often leads to tensions because then it’s one or two people chasing that goal*” *(C-03).*

Making mistakes added social pressure since they could add to the overall workload, leading to overtime for everyone. There were in all three cases tracking systems that allowed a mistake to be traced to a particular individual, which added a sense of being watched and to some degree being held accountable for the mistake. Some assemblers described this “blame culture” as a social pressure on them. Assemblers could also experience negative effects of being dependent on each other’s performance; receiving incorrect assemblies from earlier stations could add to a collective sensation of failing to carry out the task correctly, leading to everyone being negatively impacted by the need for adjustments and overtime.

#### 3.1.5. Individual Pre-Conditions and Attitudes

From the interviews, it also became obvious that most assemblers were aware that each co-worker had individual needs, assets and motivations, and that it was important to be adequately prepared, trained, equipped and motivated to assemble correctly. While many of these aspects could be viewed from the resources perspective (see Section 3.2), some intrinsic job demands were mentioned by the interviewees.

One interviewee reflected that even though they personally felt motivated to push their own performance level quite high, they were aware that “*colleagues who maybe have kids and can’t work overtime so much, they work to a different effort level and reach their targets all the same, but I like maximizing and pressuring [myself]*” *(B-21)*. Conversely, when assemblers felt decreased motivation and negative attitudes towards their work, it could result in bad assembly, characterised by “*you [doing] sloppy work, with no overview of what you’re doing. Or you work too fast and in a way that is harmful, I think*” *(C-05).*

Many spoke of different kinds of musculo-skeletal disorders that they expected to become more sensitive to with age, and certain interviewees were concerned for their younger colleagues’ future well-being: “*some (…) have worked for 15 years in assembly, which isn’t that usual (…) now that the [pace] is the way it is. But they work in a way that assures that they will last (…) they work calmly, methodically. (…) while the young are more, (…) everything has to go so fast, and preferably with a bit of movement, when fetching stuff (…) I take that extra step when I pick something up. While the young think (…) ‘I’m young, this won’t hurt me*” *(A-05).*

Several interviewees believed that the work pace was too demanding for an entire work life. This was one decisive factor for many in considering quitting their work as an assembler: “*Like I said, 15 years, you start getting tired of this. So maybe one should try something different. If you think that yeah, you’ll work until 67, it’s a long time and (…) it’s quite tough. It can be. Because it’s the line, and the tempo is high*” *(C-09).* In addition, interviewees mentioned that the work is both cognitively and physically demanding, making them tired after a working day: “*I’m tired when I’m done, but it’s physical too, not just psychological*” *(A-12)*.

Interestingly, when the assemblers were asked directly about aging aspects (Q33 and Q34), several mentioned that their current older colleagues (approximated as being between ages 40–55) appeared to be doing fine, particularly if they were experienced and had been assigned less physically strenuous stations. However, many were concerned that their own bodies would not be able to withstand the high degree of physical loading, high tempo and “running around” up until retirement age.

For others, a lack of personal interest or motivation, or wanting to reduce future exposure to the constant time pressure, was their main reason for believing they would not want to stay. Among those who believed that it would be possible and desirable to continue working until retirement in assembly, they mentioned how the “calm” and routine that comes from experience would make it possible—as long as the cycle time did not become even shorter in the future.

### 3.2. Resources in Assembly Work

Our thematic analysis identified different types of resources that facilitate work and reduce the workload for assembly workers: (i) Organisational resources; (ii) Social and interpersonal resources; (iii) Cognitive support; and (iv) Individual resources.

#### 3.2.1. Organisational Resources

The findings regarding organisational resources were divided into three sub-categories: organisation at large, organisation of work and task level.

Regarding organisation at large, motivators, onboarding process and adequate staffing were identified as the main resources (see Figure 5). The participants mentioned that their performance was influenced by their intention to continue working as an assembler, which was steered by a set of intrinsic and extrinsic motivators. The intrinsic motivators were described as the perception that the task itself is important, resulting in positive emotions, while the extrinsic motivators were described as rewards for their good work. Four motivators were identified: two intrinsic—professional pride and possibility to develop oneself; and two extrinsic—satisfying salary and favourable working hours. According to the interviewees, these motivators contributed to enhanced engagement as well as inner calm and thereby lowered cognitive demand. The intrinsic motivators were similar among the three cases. According to the interviews, professional pride is enjoying the work tasks, taking pride in assembling with quality and the feeling of being skilful when performing the work: “*It is fascinating to see what we are achieving and contributing to*” *(A-11)*. Knowing that one is good at performing one’s work also increases self-confidence and contributes to reducing nervousness of making mistakes: “*I like monotonous jobs, I am rather good at doing such things all the time, because I like it. I love working in assembly*” *(A-07).* Moreover, the possibility to further develop oneself and learn new things was mentioned to be important for work performance, increasing engagement: “*As I am very invested in my work, I think it’s really fun, and I want to learn new things*” *(A-12).* Other examples were providing the opportunity for assemblers to challenge themselves, through involvement in problem solving or receiving a promotion when showing great work performance. It was also emphasised that work created positive emotions such as joy towards their work: “*I am rather involved in all the improvement work at our department, so I am very keen to identify problems as well as solve them. I find it very enjoyable and stimulating*” *(B-01).* Regarding the extrinsic motivators, both satisfactory salary and favourable working hours were considered to influence work performance positively. As the participants were rewarded, they became more engaged in their work. Satisfactory salary was mentioned in all three cases, while favourable working hours were mentioned as a motivator in Cases B and C.

The onboarding process for introducing work tasks was further mentioned as it affects the operators’ ability to handle their work later: “*a lot is about getting a good introduction*” *(A-11)*. A common factor that was considered to influence the participants’ learning was having tutors that introduced the tasks methodically and clearly described the working steps in order to provide the assembler with sufficient time to practice: “*we should follow a certain pattern that we are confident is the smartest and smoothest way to learn (…) I believe that the most important thing is to have time to learn (…) to not feel any stress but to take it at your own pace*” *(B-19).* Having a short task sequence reduced the required amount of information to be learnt by the assemblers, which could in turn ensure product quality. The participants in Cases B and C mentioned that having the same tutor was important for continuity in their learning. In contrast, participants in Case A mentioned that having different tutors was favourable, as taking part in different operators’ working methods helped them to easily find their own working style: “*we have a standard procedure that we follow, but there are some things that are not written down exactly how to do it, since everybody has their own personal way of doing things (…) we try to have one [tutor] in the beginning and one in the end since both work differently (…) so the person learning can have the best of both*” *(A-09)*.

In all three cases, the interviewees emphasised the importance of offering all new operators the same learning conditions and expressed appreciation for when supervisors used an organised and standardised onboarding process to introduce tasks to new operators. Participants in Cases B and C also emphasised the role of a practical learning method that included assembling under the guidance of an experienced operator. Additionally, the interviewees described that giving the operators a deeper insight into the entire product and a background to why the working steps are designed in a certain way, facilitated their learning ability: “*those who have been in the main-line for a while and have seen how they build (…) have a better understanding for why we assemble things in certain ways and then also assemble better (…) in particular, they can tell when the specifications are wrong*” *(B-14)*.

Providing the operators these resources during introduction fostered skills and resulted in becoming confident in performing the work, thereby lowering cognitive load: “*I feel (…) that I could even help [others] if necessary*” *(B-01).* Besides the introduction, the participants also highlighted that they were provided with opportunities for skill development through courses such as ergonomics, material knowledge training and machine training. These were however aimed to facilitate development of selected operators with additional work responsibilities.

Adequate staffing was also emphasised by the interviewees as a resource: “*Sometimes we’re just five on the line, sometimes maybe six. It depends on how the job should be done, to have the best flow” (C-05).* Having enough staff enabled the operators to successfully perform the assigned tasks as it was described to promote workflow and thereby mitigate cognitive strain.

With regard to organisation of work (see Figure 6), three themes were identified that directly and indirectly affected cognitive demand. Participation in decision making was mentioned in all three cases as crucial, since the operators themselves often had ideas about what could optimise their workload. The interviewees highlighted different approaches for a successful involvement of the operators in decision making. One common practice was using Continuous Improvement to develop the work together with the operators: “*everything is standardised, but if we have a suggestion (…) [I] tell everybody at the meeting, ‘I have this suggestion, what do you think?’ (…) if it’s a good suggestion that saves time with no additional risks, then we go ahead*” *(A-02).* Continuous Improvements, according to the participants, consisted of regular meetings where they discussed already existing improvement proposals or wrote down ideas and thoughts that were presented orally or in writing to their managers: “*Once a month we have something called Continuous Improvement, where we write down ideas and thoughts that we later share with our (…) shift supervisor*” *(B-01).* Another common practice was communicating suggestions directly to the shift manager or writing down on a whiteboard on each line. Interviewees in Cases B and C presented their improvement proposals straight to the technicians and maintenance department. Case A assigned the operators with improvement responsibilities, which they were obliged to continuously follow up and time was set for each operator to carry out the improvement work: “*we have development targets (…) everyone should be given their time for development (…) some people are what we call brainstormers, who have a lot of ideas (…) [they] can signal while assembling and say, ‘can we do it this way?’ (…) so that one feels involved, which I think is important*” *(A-10).*

The improvement proposals were handled in different ways according to the interviewees, so were the organisations’ responses towards the proposals; for example, it was described that work instructions were developed in collaboration with the operators: “*There are binders and instructions for every workstation (…) these are formed together with our opinions. So I think these are as good as can be*” *(C-12)*. Having outdated work instructions required enlarged effort to read and contributed to confusion while assembling, therefore improving them on a regular basis could proactively diminish the risk of uncertainties. Furthermore, participants in Case A mentioned the ability to influence the sequence of the work tasks which improved performance and workflow, e.g., respondent A-09 who mentioned having occasions for improving the task sequence at the workstation as well as moving tasks between different workstations. Additionally, interviewees described that having a supportive management team and receiving feedback on their improvement proposals encouraged taking part in decision making and facilitated work performance.

Providing operators the possibility to influence the planning of work was also described by the interviewees in all three cases as facilitating the performance and reducing cognitive workload. For example, unwanted stress could be prevented by letting the operators prepare in terms of loading material and check that equipment is functional before assembling: “*I often try to start working a little before the line starts (…) If I start planning and do it now instead of later when it becomes more stressful (…) I prepare tools (…) order something that needs to be brought* via *forklift from supply*” *(B-15).* Another example mentioned in Case C was the possibility to plan work in order to successfully achieve the production goal and simultaneously reduce stress: “*sometimes we agree to work quickly, the first four hours before lunch in order to avoid stress at the end of the day. Because we have a production goal, and we need to reach it*” *(C-04).*

Manageable product variability was identified as important for creating workflow and mitigating cognitive overload. As the cases mostly had moving lines with products of a certain variation, the degree of difficulty varied and thus affected the operators’ strain: ”*I think it is good with a little variation, but when it is too much up and down it is not so good (…) when it’s a bit calmer, and then a bit harder (…) one stays active*” *(A-09).* Participants described that product variability should be distributed evenly in order to avoid peaks of demanding assembly: “*you feel how your energy drops faster if there are many difficult cars at the same time during the whole day, instead of being spread out (…) so that we don’t have to feel that stress*” *(B-13)*.

Design of the work environment was also emphasised in all three cases, as a factor that contributed to reduction of cognitive demands. Interviewees mentioned that having materials of the correct quality delivered on time was important for creating workflow and could reduce the occurrence of disturbances: “*when the material is in place, everything flows*” *(B-05).* It was also described among the participants that having material placed at the right height with the right distance to the operators’ workstations enabled easier movements: “*we had a colleague who sat and thought for a long time about how we could place the material in a way that would make it easier for us to assemble (…) it revolutionised our way of assembling*” *(A-04).* In addition, striving to have only necessary and functional equipment was found beneficial to the experience of workflow: “*when the line works as it should without technical problems, there is automatically a decent flow at the line*” *(C-05)*. 

At the task level, five factors were identified that facilitated the operator’s performance and reduced cognitive workload (see Figure 7). A well-balanced and well-planned workstation that enabled a sense of calm and the feeling of having enough time to perform the task was a good mitigator against cognitive overload. Over-balanced workstations could result in stress and operators taking shortcuts which could further lead to personal injuries and faulty assembly that reduced the quality of work. Therefore, sufficient time to complete the work was considered crucial by the participants in all three cases: “*good assembly means (…) that you have enough time. When everyone has an inner calm, when the pace is not so overwhelming that you feel that you cannot keep up (…) if the person is intelligent and starts taking short-cuts (…) short-cuts entail risks*” *(A-01)*.

According to the participants in all three cases, having as few disruptions in the work process as possible was essential for reducing cognitive workload, for example, uninterrupted work processes enabled workflow: “*I think that the work flows well when we aren’t having any stoppages*” *(B-08).* Moreover, some participants found anything outside the standard of work to be a distraction that required extra time and effort, contributing to stress: “*everything beyond the standard takes extra time and creates more stress*” *(A-14)*.

Task variety was mentioned in all three cases as an important factor when handling cognitive workload; it differed depending on what line operators were working at, and was therefore not consistent for all workstations. To avoid the feeling of monotony, variation in the tasks was vital: “*there are many different varieties (…) we perform rather many different tasks at each workstation, so we don’t get [the situation] that it becomes monotonous because you stand and do the same movements all the time*” *(A-11)*.

Frequent work rotation was described to contribute to task variation, and could additionally have a positive influence on the operators’ performance: “*good rotation, so that despite a high pace after a while, half an hour, you can do something else*” *(C-13)*. Achieving a sufficiently frequent work rotation during a work shift was considered important to uphold the assemblers’ focus and stimulation from the work itself: “*just to give the brain a different stimulus*” *(B-12).* However, the rotation frequency differed a lot, both between but also within the cases. In Case A, it was described that the rotation frequency depended on whether colleagues knew the tasks at the workstation or not: “*We try to rotate as much as possible, but not everyone knows every station, so some of us need to skip [it]*” *(A-15)*.

According to participants from Cases A and B, creating simple assembly tasks was a resource for assembly at task level. By enabling the operators to easily identify where to mount their components enhanced their work performance, as having precision demanding work tasks required heightened effort: “*the optimal way is being able to directly visually see where everything fits (…) then one doesn’t need to think about if it should be mounted here or there*” *(B-01)*. Hence, a lower degree of precision demand was essential for reducing the operators’ cognitive workload.

#### 3.2.2. Social and Interpersonal Resources

Good social relations were considered to foster enthusiasm and a positive work environment: “*Colleagues; that’s why we’re here, everyone is nice*” *(A-08).* The participants described that collegial cooperation and cohesion facilitated workflow, increased motivation and mitigated cognitive workload. Good fellowship between colleagues was described as a factor that led to good group dynamics and good communication (“*good co-workers, our group dynamic works incredibly well*” *(A-11))* that also increased motivation for being at work. The social contribution to a positive work attitude was regarded as leading to increased work performance. Participants described that they had fun while working together: “*I enjoy being with and having fun with my colleagues (…) we do our work, but also joke around*” *(B-06)*. It was also described that a well-coordinated team enabled workflow: “*it’s also that we have a good team and that everyone does exactly what they should (…) then we get a good flow*” *(C-04).*

Asking colleagues for help to complete a task that exceeded one’s ability was described as a facilitating resource when uncertainties occurred: “*when you see the component, you know that it should fit somewhere, but to double check I usually ask someone with more experience*” *(B-01).* Several interviewees stated that they preferred consulting a colleague to reading instructions: “*I’ve never had to use them, usually it just goes faster to ask someone who knows, rather than go and read it*” *(C-05).* Additionally, reminding each other in order to avoid incorrect assembly during work was mentioned as a frequent behaviour: “*I’d rather ask than make mistakes (…) people are very good at this place, so that if someone notices that you make a mistake or so, they tell you*” *(C-03)*. Moreover, some interviewees said that working faster than the pace enabled them to assist their colleagues, “*If you keep it up you can (…) finish your task and thereafter help others (…) I have a very good work team who help each other*” *(A-10)*.

Apart from receiving help from colleagues, Cases A and B had a support system called *Andon*, where the assemblers could signal and receive help from another assembler to complete work tasks: “*when you do not always remember when something appears wrong. There are so many different faults (…) we have Andon in all departments*” *(B-05).* It was also mentioned that the companies tried to have personnel resources at standby to step in and assist with demanding work tasks: “*there is either an Andon or a team leader at standby and ready to help if needed (…) we have resources that are available, so we place them in positions to support [us] if things get rough*” *(A-11)*.

#### 3.2.3. Cognitive Support

Different types of support were identified to facilitate cognitive performance and notably these were generally perceived as sufficient and satisfactory by the interviewees in all three cases: “*I often think they are very good as they are. It is very important that they exist if you are insecure*” *(B-15).* Several types of memory support were used to assist the operators in preventing incorrect assembly. This was for instance done visually by providing signals in the form of lamps to indicate if the assembler had performed the work step correctly or in the correct order: “*you can see on the computer in which step you should do [the tasks] and if you, for example, forget to tighten a screw the computer shows red. The same goes for the electric screwdriver where there are lamps that indicate if the screw is tightened or not*” *(C-03).* In addition, the visual support pick-to-light, lamps indicating which component to assemble and when, was mentioned by the interviewees to avoid both incorrect assembly as well as the risk of uncertainty while working.

In Cases A and B, the participants described that markings such as arrows or colours were used to facilitate assembly: “*we have screws of different length, and they are colour coded*” *(A-09)*. Another memory support mentioned in Cases A and C was a *quality alarm*, used to detect incorrect or unperformed assembly tasks by hindering the moving line, thus preventing the operators from continuing assembly: “*a system with detectors, so that you cannot go ahead with the sequence if you don’t finish a certain task, so that way they guarantee that we assemble correctly*” *(A-04)*.

Each case used different kinds of memory support solutions that were unique to them. In Case A, scanners were used to help remembering: “*we have a scanner, so if you use it, it’s impossible to forget*” *(A-15).* Another memory support used in Case A was a type of screen technology that also reminded the assembler and facilitated their cognitive performance: “*[screens] that tells you when you are finished by turning green. Then the task is checked off and the line continues*” *(A-08, A-10).* Moreover, receiving pre-kitted boxes with materials was an appreciated resource in Case B. According to participant B-10 the components were marked and placed correctly in their carts which enabled workflow during assembly, both in terms of knowing how to place the components and ensuring that the correct number of components were mounted. It was also mentioned by the interviewees in Case B that computers were used to assist in detecting electrical errors: “*We have a computer that scans for electrical problems, so often (…) we get a printed document from our colleagues at line showing what is wrong*” *(B-06)*. Vision cameras were used to assist in finding errors and alert the personnel in Case C. According to the participants these could detect most of their errors, thus this was perceived to be an essential resource to prevent incorrect assembly: “*The memory aids are the cameras, I guess (…) it is very difficult to get anything wrong because we have these cameras (…) that control*” *(C-08)* and “*I think it identifies faults by vision up to 98%*” *(C-13, C-15)*.

Another cognitive support was work instructions, presented at each workstation on paper in a binder and sometimes also digitally through a computer or screen to refresh the assemblers’ memory. Having work instructions contributed to ensuring that operators followed the standards, and therefore it was highlighted by the interviewees as an important resource: “*We have everything documented in binders at each station, describing the work method that we follow (…) it doesn’t matter if there are ten different assemblers, they should all work the same way. That is how we are trained*” *(A-03).* The interviewees also described that work instructions should be precise and easy to read to facilitate usage and upholding workflow. All cases had several kinds of instructions where the level of detail varied as well as the format in terms of visualization and text: “*there is a binder with different standardisations on what to do and how (…) there are pictures of the models from above, from below, from the side*” *(C-06).* Moreover, it was emphasised that the instructions were developed to accommodate even inexperienced assemblers to ensure correct assembly: “*it should be possible for people who don’t know how to read these and be able to build afterward*” *(B-16)*.

#### 3.2.4. Individual Resources

The individuals themselves could also influence their ability to handle the workload: “*If you can work more efficiently, you do it*” *(C-01)*. Level of experience was mentioned in all three cases as impactful on work performance; interviewees expressed how feeling relaxed and being skilful while assembling contributed to workflow: “*I very seldom feel that I have to hurry. This has a lot to do with experience (…) one feels rather relaxed (…) you know exactly where everything is. I can almost stand there and close my eyes and assemble nowadays*” *(A-06)*. Being focused and having the mindset of trying to assemble with as high quality as possible was another common factor affecting the assemblers’ performance positively: “*A good assembly, that is essentially doing your best in the time you’ve got*” *(B-09)*. Participants in all three cases also described that taking time for self-checks enabled them to avoid further adjustments requiring extra effort: “*you should check the car after assembling (…) then you can find those errors*” *(B-19)*.

Memory training was further identified as facilitating the assembling performance as it mitigated cognitive overload: “*When I stand at the workstation, I no longer think, I just work (…) As I have trained my muscle memory and it sticks*” *(A-03)*. Additionally, interviewees mentioned that finding routines enabled the operators to incorporate their skills: “*Foremost you find routines. That this is my way of working at this workstation and then you always follow the same pattern, and then it becomes part of your muscle memory*” *(C-05)*. Interviewees in Cases A and B highlighted that writing down notes was a helpful method to avoid forgetting things: “*I also have a note-book, otherwise you cannot remember everything*” *(B-03)*.

## 4. Discussion

The aim of this study was to gain an increased knowledge about assembly workers’ cognitive/mental workload conditions. Based on a qualitative multi-case study, this paper has mapped contextual job demands and resources that influence *cognitive/mental* workload, well-being and performance in assembly work from three industrial cases. The identified demands as well as resources are distributed at different levels within the entire manufacturing system, from organisational design down to individual worker characteristics. The differentiations found between job demands and resources were mostly compatible with the sub-categorizations in Bakker and Demerouti’s model [29].

### 4.1. Reflections on Findings and Literature Comparison

Despite much of their focus being on physical demands, we find that industrial assemblers face a variety of cognitive/mental demands when performing their assembly work. Several of our findings echo previous literature—often in relation to outcomes on quality—such as the effects of high product assembly complexity on corrective adjustment costs [1,49], the combination of alertness (as a function of the time of day) and task complexity combined with time pressure [50], presentation of materials affecting time pressure and quality [16], and conditions for on-boarding and on-the-job training, including a combination of instructions and learning assembly alongside an experienced colleague [17].

Interestingly, in another interview study with experienced assembly operators, “psychologically demanding” tasks (where the wording was chosen in accordance with the term used by interviewees) were associated with product quality deficiencies, in that “*an important factor for job satisfaction was the possibility for the workers to perform their tasks with high quality”* [51] (p. 15), and that assemblers felt irritation when components were a bad fit—all of these factors are mentioned in our study, and appear confirmed. A social sustainability KPI instrument by Scafà et al. [15] also covers a number of our identified cognitive demands and resources from individual, organisational and workplace design perspectives, labelling demands as “risks” (e.g., Precise and fine movements; Task variety; Task repetitiveness; Work rhythm; Shift management; Decision-making autonomy; Work relationships; Lack of skills; Lighting; Noise; Functionality of work equipment; Lacking or complex work instructions) and resources as “corrective actions” (e.g., Optimize the workflow; Workload balancing; Limit giving workers tasks that under-utilize their skills; Rotate tasks and schedules; Give workers some control over the way they do their work; Training programs according to workers’ skills; Pairing with expert workers in the event of new roles/tasks; Provide best-practices to better execute tasks).

The results indicate that the participants, when prompted, perceived many different resources that influenced their cognitive performance. Remarkably, all three case companies provided ample resources that reduced the cognitive workload, even though the companies’ intentions were mostly to increase assembly quality and reduce errors.

Towards a similar end goal, shortened task sequences were intended by companies to reduce the demand of having to learn and memorise an enlarged amount of information, but by only allowing the assemblers to know a selected part of the entire product, their overall picture and understanding of why to assemble in a certain way became limited. This constitutes a goal conflict between decreased memory load and enabling an understanding of the entire product. Earlier findings [1,49] state that handling several complex operations is cognitively demanding and aggravates the failure rate; thus, the strategy of shorter sequences appears partially supported in the literature (as this demands less memorisation). Despite the cases’ differences in terms of corporate culture and company maturity (e.g., with regard to offering formalised training), most types of provided resources were similar. Workplace design and tool resources tended to share similar intended outcomes, although they varied in principle and technology type.

The findings indicate interrelations between cognitive loading and physical aspects, as well as assemblers’ ideas about the physicality of the work, both in the short and long term. Physically demanding tasks without sufficient ergonomic support seemed to lead to both exhaustion and depletion of cognitive resources. Social resources could help buffer the cognitive workload, while work-environmental and organisational demands, such as noise, sub-optimal workstation design and inadequate staffing, training or instructions increased cognitive demands. Thus, to examine what affects cognitive performance in industrial assembly, a holistic consideration of working conditions is required.

Several job demands and resources are tightly interlinked, to the point of often being two sides of the same coin: if a particular type of resource is deemed insufficient or missing, its absence often constitutes a perceived (cognitive/mental) demand. Conversely, certain demands were perceived by some interviewees as positive performance challenges. This response relies mainly on the individuals’ resources and attitudes. This reflects a known conundrum with the JD-R model: a negatively appraised resource may be seen as a threat and can be classified as a demand, while a positively valued demand can be seen as an appreciated challenge, and thereby classified as a resource [52].

In cases where individual experience and capability is the main resource that allows coping with demands, we see a risk that companies may enter a slippery slope and rely too much on recruiting a certain type of personnel that fits in and adapts to challenges, instead of striving for diversity and discovering improvement potentials. To achieve an inclusive, equitable work environment, it is important that companies design environments to fit individuals with different abilities and pre-conditions, and avoid having environments that only match particular abilities and coping strategies.

### 4.2. Methodological Quality Assurance

A strength of our study is the triangulation between three different companies with varying levels of production complexity and different organisational cultures. Therefore, we were able to establish a broad picture of the work conditions in assembly work, and its specific demands and resources with a focus on cognitive performance.

Several strategies were used to ensure the quality of this qualitative study. First, the research results were discussed iteratively and thoroughly within the research group throughout the analysis to avoid biases [53]. One of the aspects that we resolved during these discussions was with respect to the JD-R model as mentioned above. Second, the findings were presented to representatives at each of the companies to confirm the results, complement missing issues and avoid misinterpretations (cf. [53]). Third, to ensure the trustworthiness of our findings, we provided rich descriptions and direct quotes from our data [53]. Our study employs several of the design strategies and techniques listed by Riege [54] to establish qualitative case study validity and reliability, such as (but not limited to) Establishment of a chain of evidence; Reviewing of drafts by case company representatives; Within- and cross-case analysis; Definition of scope and boundaries in the design phase; Use of multiple researchers; Mechanical data recording; and Peer debriefing ([54] pp. 82–83).

### 4.3. Study Limitations

Some limitations of our study are discussed here. Firstly, the purpose of carrying out interview-based qualitative studies is to provide a deeply context-specific, rich account of each interviewee’s perspective of a situation [55]. Therefore, some degree of subjectivity is inherent in the method [44,56], particularly when using interview techniques to understand the psychological reality of others [57]. Some influences can also stem from the researchers themselves; we therefore completed a COREQ checklist [42] to clarify the research team’s background, experience and research interests, to make transparent our discipline-related and theoretical point of departure.

Although we deliberately focused the interview questions (and hence this paper) on cognitive aspects, the participants clearly described assembly work as “more physical than mental” and recognised physical loading as the main source of strain. Although the original JD-R model includes physical aspects, we delimited these from our reporting and used the model as a framework to organise the purely cognitive/mental aspects of assembly work. However, since the awareness of physical loading is a significant part of the mental experience of assembly work, we want to emphasise that its influence remains very important.

The participants seemed to have a limited knowledge of what constituted the cognitive aspects of assembly work and used a limited, informal vocabulary to approach this complex phenomenon. Furthermore, since the mental workload concept (in literature) implies that the physical and cognitive sensations of work interact with each other, it seemed consistent that assemblers struggled to clearly distinguish between the two. This tendency had been anticipated following early-stage conversations with the contact person at each company, where the issue of how to speak about something as intangible as cognitive ergonomics had been brought up. This resulted in a study design where, rather than asking generally about “cognitive loading”, the questions in the semi-structured interview guide (Appendix A) targeted more tangible issues (informed by literature studies) of cognitive/mental loading in manufacturing contexts, e.g., memory demands, distractions, teamwork and frustration. In this way, the interview results were able to sidestep any potential lack of understanding of the concept cognitive/mental loading.

Due to practical study constraints, which limited the time available for speaking to company employees, our interview sample did not target managers and supervisors (or, indeed, management and leadership questions). Interviewing these roles could have enriched the study, particularly regarding the organisational aspects of assemblers’ resources for handling cognitive/mental demands.

### 4.4. Practical Implications

A clear implication from this study is that no single role or stakeholder in a manufacturing system bears the sole responsibility to address the problem of unsuitable cognitive/mental workload; because the problem is systemic, so is the solution, meaning that improvement potentials exist at both micro- and macro levels, and that resilience of the entire system can be reinforced both with individual-focused interventions, such as training and skills development, as well as through systemic characteristics, such as design solutions in the workplace, organisational structures and beneficial collegial interactions. The responsibility for identifying demands at various levels in the organisation and addressing them with improvements or interventions therefore appears distributed across many different roles. Early phases of product and production design, where design and manufacturing engineers are involved, are considered to be the most optimal and cost-effective stages for proactive ergonomics action [58,59]. However, proactive interventions in production have predominantly been considered from a physical perspective, and less so from a cognitive/mental loading perspective. The mapping of demands and resources provided in this paper can be used both for reactive and proactive intervention purposes. Specifically, the results can be used as a risk assessment guide with a focus on cognitive ergonomics, applied for evaluation of both existing assembly lines and production concepts in design processes. 

Rather than try to balance all cognitive /mental workload demands at one particular phase in the product/production development process, we argue that companies can benefit from a broader view of what a cognitive resource can be, and who or what can supply it. For example, recognizing that clear and easy-to-read instructions are an important resource may result in a more assembler-centred focus when tasking manufacturing engineers with creating instructions, while recognizing social collegial interactions as a resource for assembling well might be valuable for a Human Resources role to reinforce. 

Furthermore, the unsuitable cognitive/mental workload problem is itself a moving target within a highly adaptive socio-technical system: although individual assemblers may encounter demands, they often find individual or collective strategies and “tricks of the trade” to meet those demands. However, relying entirely on individuals’ capacity to cope with demands may be a slippery slope: companies should avoid the detrimental trade-off where assemblers sacrifice their long-term well-being, safety, engagement and motivation to “get the job done” in the short term. Their perseverance coupled with dissatisfaction appears conducive to wanting to leave the profession as an assembler, which is a cost and a knowledge loss for companies. 

Finally, it appears wise to not completely separate the physical from the cognitive/mental workload; although this paper has adopted a very intentional focus on mental aspects to give it dedicated attention, re-incorporating the interactions between the physical and mental experience of assembly work remains a highly relevant endeavour (as elaborated by, e.g., the authors in [7,60,61,62,63]).

### 4.5. Future Work

Some directions for future work using the present results could include deeper exploration of some of the emergent themes, a cross-case analysis, and possibly looking closer at how case characteristics such as company size, complexity of the products or workstations, the procedure of onboarding/training of new assemblers, etc., could have influenced the results. Our present study refrains from quantifying the exact numbers of respondents who provided statements towards a particular theme, since reporting under the JD-R framework is already quite complex and lengthy (and because interviewees responded to the questions at very different degrees of detail). A pure cross-case analysis could be a good opportunity to draw on the explanatory power of mentioning the numbers of respondents adding insights to each theme; however, counting in qualitative studies is not entirely uncontroversial [64].

Another possibility would be the creation of a method, guide or checklist tool from this more holistic perspective. Many different methods for mapping cognitive and/or mental workload already exist (e.g., [13,19,65,66,67,68,69,70,71]), but the vast majority of them avoid combining multiple demand/resource factors with a multi-level (micro to macro) responsibility perspective. It would in that case be crucial to define which roles in a manufacturing organization are to be involved in the use of the tool, as the broad scope of what would be considered a cognitive demand and/or resource could befall very different areas of responsibility. Nevertheless, an organisation-wide coordinated effort to recognise and address cognitive/mental workload issues could increase general awareness of the importance of the topic, as well as reinforce assembler willingness to stay on in the profession. 

On a research-related note, mapping the different demands and resources is a first step towards more advanced analyses of how these system elements interact; an interesting avenue going forward might be to employ *Necessary Condition Analysis* [72] to try and define which of the identified demands and resources are necessary and sufficient (present or absent) conditions for a particular desirable or undesirable outcome. This methodology has been used in a manufacturing context before [73]. Such a study would for our purposes require additional data collection and a new study design. One caveat towards this idea may be that some desirable states of “cognitive resilience” from the assembler perspective may be on a continuum, and vary with regard to the individual’s skill development, rather than a question of presence/absence. However, the more systemic and organisational aspects might be worth exploring, as this could be a step towards justifying more impactful system interventions to optimise cognitive/mental workload. A further development from there-on might involve the verification of related constructs using e.g., surveys (to explore possible directional relationships between demands, resources and outcomes) followed by Exploratory or Confirmatory Factor Analysis.

## 5. Conclusions

In this study, demanding work conditions and facilitating factors were identified in industrial assembly work and results were organised using the JD-R model. The categorisation of demands and resources at different levels within an organisation, from the organisation at large to the individual level, clarifies that achieving a well-balanced workload cannot be handled at one organisational level in isolation, but requires collaboration and action within the entire organisation.

A clear connection between demands and resources was emphasised during the interviews as creating proper conditions for the assemblers while working; demands and resources are often two sides of the same coin. A well-thought-out setup, good group cooperation and stimulating work tasks are some of the resources that can mitigate cognitive over- or underload, whereas disturbances, dysfunctional group dynamics and unbalanced work tasks sub-optimise the cognitive load.

Therefore, we propose that a sustainable level of cognitive workload in assembly work is a state of balance between different types of (i) work *demands* that keep assemblers alert and engaged with their cognitive work tasks, matching their skill level and maintaining a sense of control; and (ii) work *resources* that provide the necessary training, instruction, tools and support from colleagues and supervisors. A sustainable cognitive workload should enable assemblers to perceive relevant signals from the assembly situation; recognise, process and interpret them; and make timely and appropriate decisions for action, despite daily variations with respect to mental loading factors such as emotional state, frustration levels and even the time of day. This knowledge should enable companies as a whole to provide optimal working conditions for assembly workers.

## Figures and Tables

**Figure 1 ijerph-18-12282-f001:**
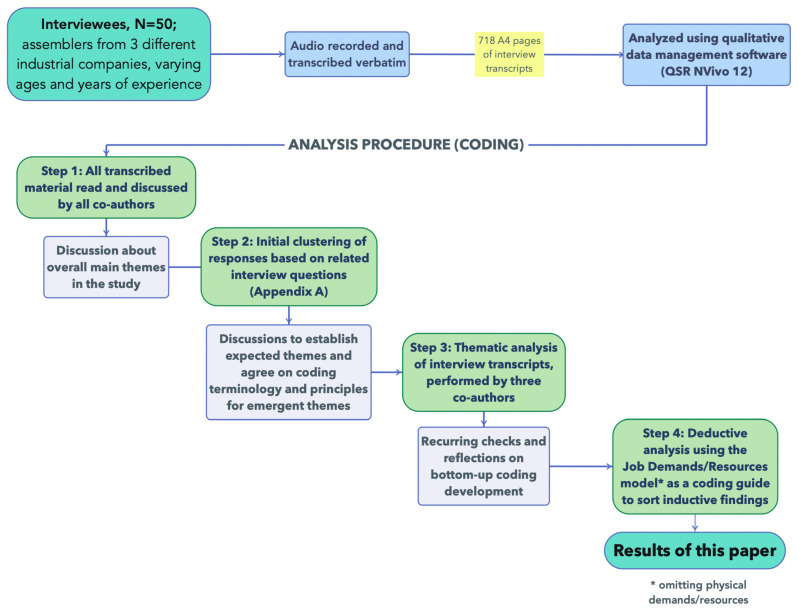
The data collection and analysis process.

**Figure 2 ijerph-18-12282-f002:**
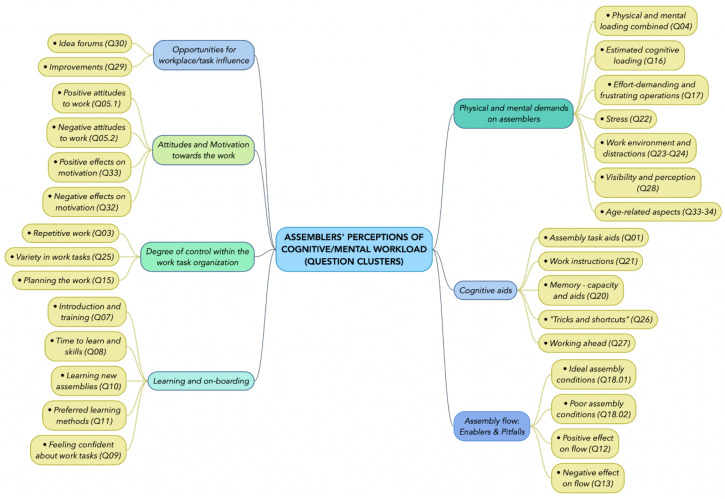
The initial “clustering” of related questions (see full questions in Appendix A).

**Figure 3 ijerph-18-12282-f003:**
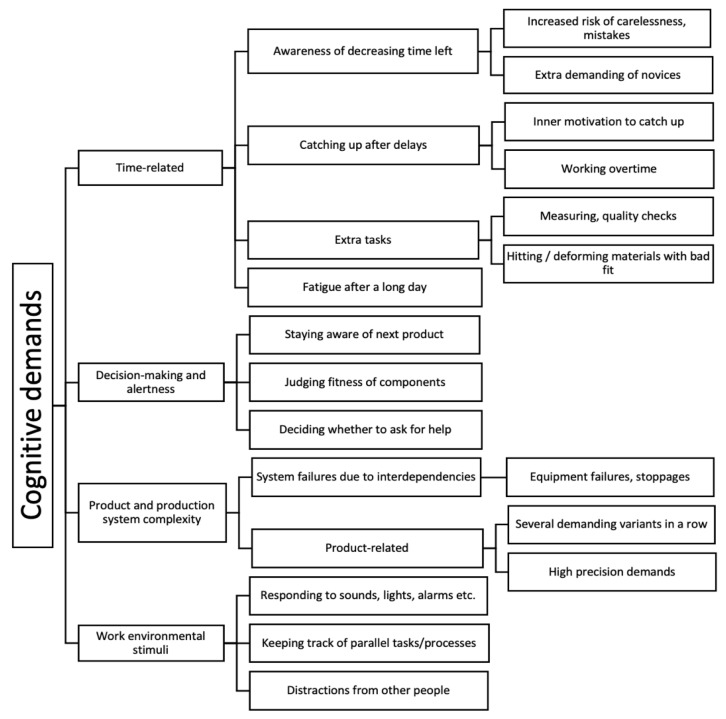
Cognitive demands.

**Figure 4 ijerph-18-12282-f004:**
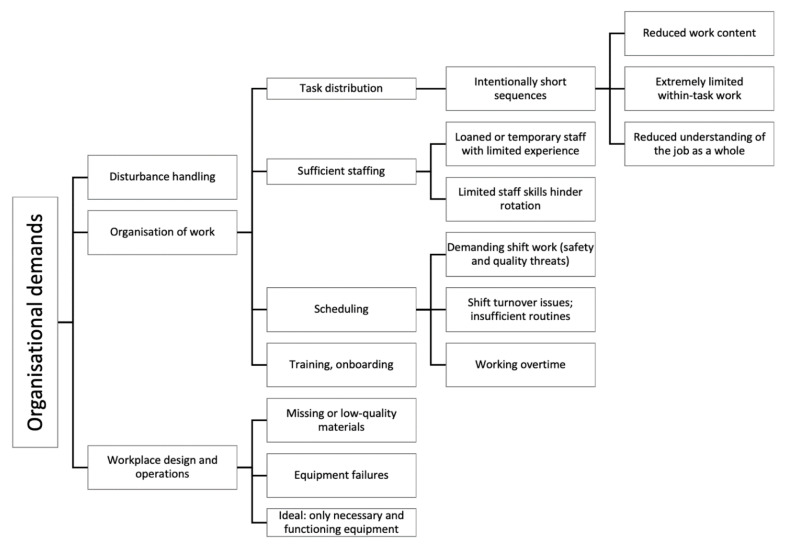
Organisational demands.

**Figure 5 ijerph-18-12282-f005:**
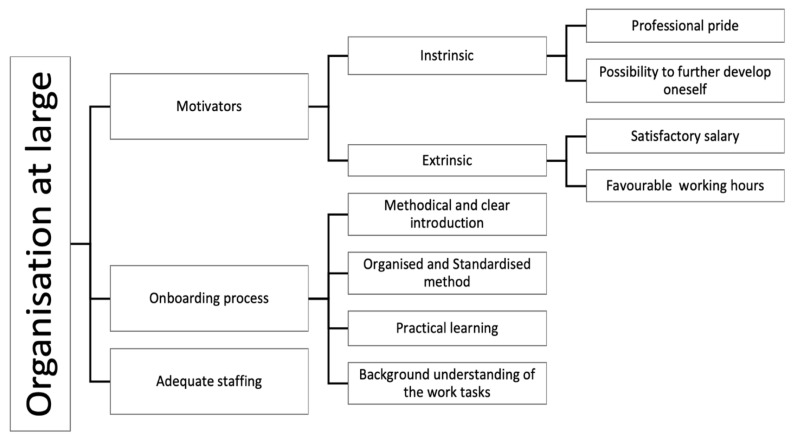
Resources regarding organisation at large.

**Figure 6 ijerph-18-12282-f006:**
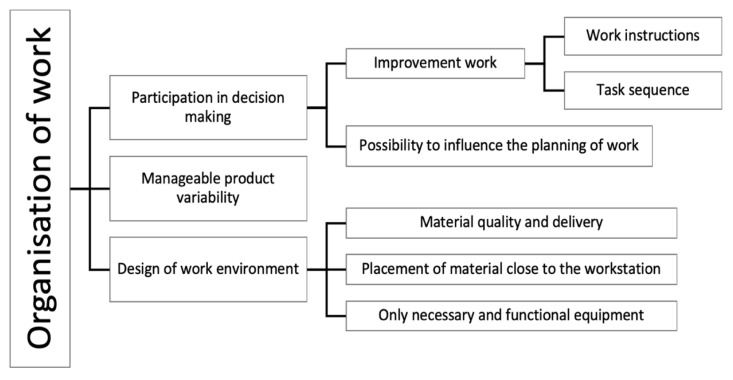
Resources regarding the organisation of work.

**Figure 7 ijerph-18-12282-f007:**
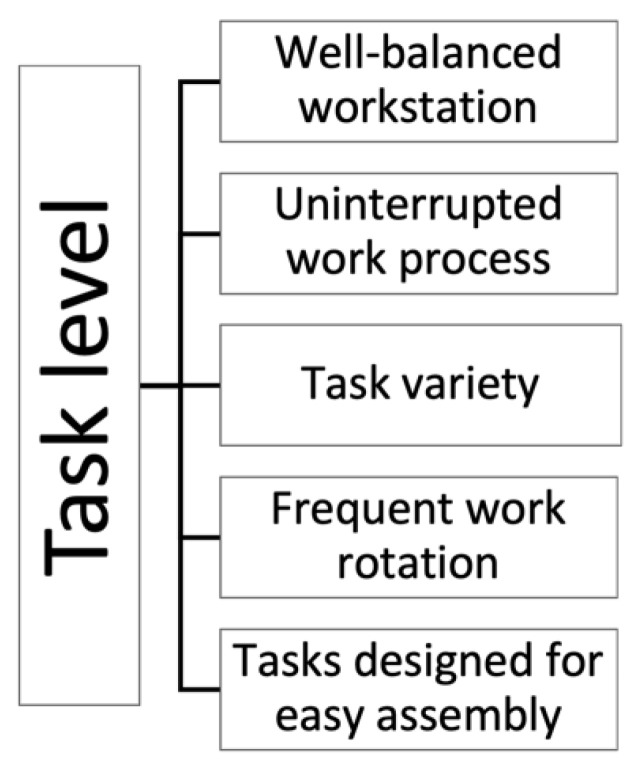
Resources at the task level.

**Table 1 ijerph-18-12282-t001:** Demographic information about the assemblers (total *n* = 50).

Company	Gender Distribution (M = Male, F = Female)	Age Span	Work Experience Span
A (vehicle manufacturer), *n* = 15	10 M, 5 F	22–54 yrs	10 mo–32 yrs
B (vehicle manufacturer) *n* = 22	16 M, 6 F	20–56 yrs	5 mo–39 yrs
C (automotive component manufacturer), *n* = 13	5 M, 8 F	19–60 yrs	6 mo–30 yrs

## Data Availability

The data presented in this study are available, in the form of anonymized transcripts in the original language, on request from the corresponding author. The data are not publicly available due to the risk of interview citations out of context, and to protect the identities of mutually recognisable co-workers (the context of the study may enable one interviewee to recognise another).

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
