# Peer review of "Cognitive Ergonomics of Assembly Work from a Job Demands–Resources Perspective: Three Qualitative Case Studies"

_ijerph, 2021, doi:10.3390/ijerph182312282_

Round 1

Reviewer 1 Report

I have gone through the review and I think that my comments have been met and have nothing further to add. I think that the paper is ok now!

Author Response

Thank you for very helpful comments that helped us improve the paper! 

Reviewer 2 Report

The authors have responded appropriately to the reviewers' suggestions.

Author Response

Thank you for very helpful comments that helped us improve the paper! 

This manuscript is a resubmission of an earlier submission. The following is a list of the peer review reports and author responses from that submission.

Round 1

Reviewer 1 Report

A really interesting work!

A few comment and clarifications:

  • Appendix A is missing in the manuscript.
  • On row 109: it should be “ mapping”.
  • On row 173: It states “Figure #”, it should be “Figure 1”.
  • It would be interesting to know if you could see any difference of the results in the different cases (big vs. small company) or a how “complex/ heavy” workstations influenced the results. This may not fit into this paper, but it would be nice if you could make a short reflection of this or perhaps a comment in future work.
  • It would also be interesting to include some more quantities in order to understand the impact of some of the data. Perhaps something for future more deeper analysis.

Reviewer 2 Report

The core of this study is to create a taxonomy. However, the purpose of taxonomy development is unclear. If it is really for evaluation, the hierarchy should be a little shallower, and reliability and validity should be verified. But the most important part of validation is missing. If the authors want to secure procedural justification, the authors should describe the procedure in more detail. Or, it is better to conduct EFA and CFA for such types of models. Only subjective findings seem to be weak.

The authors consciously excluded physical demand and explained it. But at the same time, the authors tried to cover the overall demand by adding an organizational demand. It would be nice if I could agree with this, but I don't. I think physical demand should also be actively included. If numerical evidence was supported, I might have had to agree. I feel that the subjectivity of the authors is too much involved in this study.

It is regrettable that a lot of input from managers and supervisors is not included in the conceptualization process, such as organizational demand, and this reduces the completeness of the result.

Reviewer 3 Report

The paper provides a very important set of data on the job and resource demands of assembly workers. These results are analysed within the framework of JD-R theory.

I consider that the paper should be accepted for publication in the journal. However, I have some questions and suggestions to make.

Firstly, in the introduction there is a reference to the distinction between cognitive load and mental workload that I do not think is well explained. These two concepts come from two different fields of work whose interests and perspectives have led to two definitions that have been confused and confusing for many years. Mental workload comes from the field of Ergonomics and Human Factors and refers to the discrepancy between the demands of the job and the ("mental") resources available to the worker. In this field, attention has only recently been paid to the "effort" that workers put into the task. For ergonomists and Human Factors specialists, the worker puts all the resources he/she has to cope with the demands of the task. However, from the other field of research, educational psychology, the effort is not always assumed, and the focus is on whether the person puts all the resources available to perform the task. This is the reason for preferring the term “cognitive load” among educational psychologists.

I think this is the big difference between the two concepts. But the authors do not make this difference clear in the introduction and the reader is left confused. For this reason I would ask that the authors rewrite the introduction to clarify these concepts.

Regarding the methodology, I think it is correct, but I would appreciate it if the interview questions were explained. The reader does not know to what questions the workers answered.